# Women in the European Virus Bioinformatics Center

**DOI:** 10.3390/v14071522

**Published:** 2022-07-12

**Authors:** Franziska Hufsky, Ana Abecasis, Patricia Agudelo-Romero, Magda Bletsa, Katherine Brown, Claudia Claus, Stefanie Deinhardt-Emmer, Li Deng, Caroline C. Friedel, María Inés Gismondi, Evangelia Georgia Kostaki, Denise Kühnert, Urmila Kulkarni-Kale, Karin J. Metzner, Irmtraud M. Meyer, Laura Miozzi, Luca Nishimura, Sofia Paraskevopoulou, Alba Pérez-Cataluña, Janina Rahlff, Emma Thomson, Charlotte Tumescheit, Lia van der Hoek, Lore Van Espen, Anne-Mieke Vandamme, Maryam Zaheri, Neta Zuckerman, Manja Marz

**Affiliations:** 1European Virus Bioinformatics Center, 07743 Jena, Germany; ana.abecasis@ihmt.unl.pt (A.A.); patricia.agudeloromero@telethonkids.org.au (P.A.-R.); magda.bletsa@kuleuven.be (M.B.); kab84@cam.ac.uk (K.B.); claudia.claus@medizin.uni-leipzig.de (C.C.); stefanie.deinhardt-emmer@med.uni-jena.de (S.D.-E.); li.deng@helmholtz-muenchen.de (L.D.); caroline.friedel@bio.ifi.lmu.de (C.C.F.); gismondi.maria@inta.gob.ar (M.I.G.); ekostakh@med.uoa.gr (E.G.K.); kuehnert@shh.mpg.de (D.K.); urmila@bioinfo.net.in (U.K.-K.); karin.metzner@usz.ch (K.J.M.); irmtraud.meyer@mdc-berlin.de (I.M.M.); laura.miozzi@ipsp.cnr.it (L.M.); rnishimura@nig.ac.jp (L.N.); paraskevopoulous@rki.de (S.P.); alba.perez@iata.csic.es (A.P.-C.); janina.rahlff@lnu.se (J.R.); emma.thomson@glasgow.ac.uk (E.T.); ct518@cam.ac.uk (C.T.); c.m.vanderhoek@amc.uva.nl (L.v.d.H.); lore.vanespen@kuleuven.be (L.V.E.); annemie.vandamme@kuleuven.be (A.-M.V.); zaheri.maryam@virology.uzh.ch (M.Z.); neta.zuckerman@sheba.gov.il (N.Z.); 2RNA Bioinformatics and High-Throughput Analysis, Friedrich Schiller University Jena, 07743 Jena, Germany; 3Global Health and Tropical Medicine, Institute of Hygiene and Tropical Medicine, New University of Lisbon, 1349-008 Lisbon, Portugal; 4Wal-Yan Respiratory Research Centre, Telethon Kids Institute, University of Western Australia, Nedlands, WA 6009, Australia; 5Department of Hygiene, Epidemiology and Medical Statistics, Medical School, National and Kapodistrian University of Athens, 115 27 Athens, Greece; 6Department of Microbiology, Immunology and Transplantation, Rega Institute, Katholieke Universiteit Leuven, B-3000 Leuven, Belgium; 7Division of Virology, Department of Pathology, University of Cambridge, Cambridge CB2 1TN, UK; 8Institute of Medical Microbiology and Virology, Medical Faculty, Leipzig University, 04103 Leipzig, Germany; 9Institute of Medical Microbiology, Jena University Hospital, 07747 Jena, Germany; 10Institute of Virology, Helmholtz Centre Munich-German Research Center for Environmental Health, 85764 Neuherberg, Germany; 11Microbial Disease Prevention, School of Life Sciences, Technical University of Munich, 85354 Freising, Germany; 12Institute of Informatics, Ludwig-Maximilians-Universität München, 80333 Munich, Germany; 13Institute of Agrobiotechnology and Molecular Biology (IABIMO), National Institute for Agriculture Technology (INTA), National Research Council (CONICET), Hurlingham B1686IGC, Argentina; 14Department of Basic Sciences, National University of Luján, Luján B6702MZP, Argentina; 15Transmission, Infection, Diversification and Evolution Group, Max Planck Institute for the Science of Human History, 07745 Jena, Germany; 16Bioinformatics Centre, Savitribai Phule Pune University, Pune 411007, India; 17Department of Infectious Diseases and Hospital Epidemiology, University Hospital Zurich, 8091 Zurich, Switzerland; 18Institute of Medical Virology, University of Zurich, 8057 Zurich, Switzerland; 19Berlin Institute for Medical Systems Biology, Max Delbrück Center for Molecular Medicine in the Helmholtz Association, 10115 Berlin, Germany; 20Institute of Chemistry and Biochemistry, Department of Biology, Chemistry and Pharmacy, Freie Universität Berlin, 14195 Berlin, Germany; 21Faculty of Mathematics and Computer Science, Freie Universität Berlin, 14195 Berlin, Germany; 22Institute for Sustainable Plant Protection, National Research Council of Italy, 10135 Torino, Italy; 23Department of Genetics, School of Life Science, The Graduate University for Advanced Studies (SOKENDAI), Mishima 411-8540, Japan; 24Human Genetics Laboratory, National Institute of Genetics, Mishima 411-8540, Japan; 25Methods Development and Research Infrastructure, Bioinformatics and Systems Biology, Robert Koch Institute, 13353 Berlin, Germany; 26VISAFELab, Department of Preservation and Food Safety Technologies, Institute of Agrochemistry and Food Technology, IATA-CSIC, 46980 Valencia, Spain; 27Centre for Ecology and Evolution in Microbial Model Systems (EEMiS), Department of Biology and Environmental Science, Linneaus University, 391 82 Kalmar, Sweden; 28Queen Elizabeth University Hospital, NHS Greater Glasgow and Clyde, Glasgow G51 4TF, UK; 29MRC-University of Glasgow Centre for Virus Research, Glasgow G61 1QH, UK; 30School of Biological Sciences, Seoul National University, Seoul 08826, Korea; 31Laboratory of Experimental Virology, Department of Medical Microbiology and Infection Prevention, Amsterdam UMC, University of Amsterdam, 1012 WX Amsterdam, The Netherlands; 32Amsterdam Institute for Infection and Immunity, 1100 DD Amsterdam, The Netherlands; 33Global Health and Tropical Medicine, Instituto de Higiene e Medicina Tropical, Universidade Nova de Lisboa, 1349-008 Lisbon, Portugal; 34Institute for the Future, Katholieke Universiteit Leuven, B-3000 Leuven, Belgium; 35Central Virology Laboratory, Public Health Services, Ministry of Health and Sheba Medical Center, Ramat Gan 52621, Israel

**Keywords:** virus bioinformatics, big data, networking, virus discovery, virus evolution, viral infection, transcriptomics, emerging viruses, epidemiology, viral ecology

## Abstract

Viruses are the cause of a considerable burden to human, animal and plant health, while on the other hand playing an important role in regulating entire ecosystems. The power of new sequencing technologies combined with new tools for processing “Big Data” offers unprecedented opportunities to answer fundamental questions in virology. Virologists have an urgent need for virus-specific bioinformatics tools. These developments have led to the formation of the European Virus Bioinformatics Center, a network of experts in virology and bioinformatics who are joining forces to enable extensive exchange and collaboration between these research areas. The EVBC strives to provide talented researchers with a supportive environment free of gender bias, but the gender gap in science, especially in math-intensive fields such as computer science, persists. To bring more talented women into research and keep them there, we need to highlight role models to spark their interest, and we need to ensure that female scientists are not kept at lower levels but are given the opportunity to lead the field. Here we showcase the work of the EVBC and highlight the achievements of some outstanding women experts in virology and viral bioinformatics.

## 1. Introduction

Emerging and existing viruses continue to be a pressing issue in the face of an ever-growing and aging global population, posing major challenges to health science, technology and society at large. In recent years, we have witnessed the emergence of new viral diseases (e.g., COVID-19), the re-emergence of known diseases in new geographical areas (e.g., dengue), the increasing socio-economic burden of a wide range of lower virulence viruses (e.g., herpes viruses), and viral diseases in animals and plants that affect security of food supply. In addition, there are a staggeringly large number of viruses in the biosphere, of which only a tiny fraction have been identified [1]. In view of this, ICTV, (International Committee for Taxonomy of Viruses), has re-initiated discussions on the need to catalog and classify millions of viruses that are expected to be discovered through metagenomics/viromics initiatives on the basis of their genomic sequences alone [2,3]. While some viruses are pathogens, other viruses can be repurposed for therapies and a vast majority of viruses can play an important role in regulating entire ecosystems.

The most urgent open-ended questions in fundamental virus research require dedicated methods to be answered and can only be addressed with the help of bioinformatics. Recent biotechnological developments (such as high-throughput sequencing or metagenomics) have brought virology into the age of big data. Researchers can be overwhelmed by the sheer volume of data, and traditional methods of interpretation are prone to error. Compared to e.g., bacteriology, the required sophisticated bioinformatic tools for the analysis of virus data are only recently developed.

These developments have led to the formation of a network of experts in virology and bioinformatics who are joining forces to enable extensive exchange and collaboration between these research areas. Here we present the engagement of the European Virus Bioinformatics Center and highlight the achievements of some outstanding female experts in virology and bioinformatics.

## 2. The European Virus Bioinformatics Center

The European Virus Bioinformatics Center (EVBC) was founded in 2017 and currently has 247 members (see Figure 1) from 140 research institutions distributed over 36 European and non-European countries. The center is bringing together the excellence of virology and bioinformatics from different sectors, domains, disciplines and background into a unique consortium. It aims at solidifying the exchange of ideas, initiating scientific cooperation between bioinformaticians and virologists and increasing the international visibility of virus bioinformatics. In addition, we promote young scientists and advance teaching of virus bioinformatics. If you are interested in joining the EVBC, please fill in the application form (http://evbc.uni-jena.de/membership-application/).

The EVBC is engaging in several activities to reach these aims (see Table 1 for links to all services). A detailed overview about our work can be found on the EVBC website (https://evbc.uni-jena.de/ (accessed on 24 May 2022)). We are publishing a monthly newsletter, informing about recent research results, upcoming events, job vacancies and further announcements, and also have a very active Twitter community spreading recent research (see Section 2.1). We are organizing an annual conference on virus bioinformatics (see Section 2.2), as well as several monthly lecture series (see Section 2.3) and workshops (see Section 2.4). We are curating a list of bioinformatics tools to be applied in virology (see Section 2.5). We implement collaborative jointly funded projects on bioinformatics and virology that achieve more than the sum of their parts (see Section 2.6). We are regularly editing special issues on virus bioinformatics in different scientific journals (see Section 2.7).

During the SARS-CoV-2 pandemic, the EVBC was supporting the researchers by collecting newly developed tools designed for fast detection and understanding of SARS-CoV-2, and treatment of COVID-19 [4] and by keeping people up-to-date about recent research findings.

### 2.1. Stay up to Date: EVBC Newsletter, Calendar and Twitter

One of the main services of the EVBC is our monthly newsletter to keep people informed about recent research results, upcoming events, job vacancies and further announcements. We are constantly monitoring the publications of our members and highlight all virus bioinformatics related papers in the newsletter. Newly developed tools are added to our tools collection (see Section 2.5). We are curating an event calendar covering virus bioinformatics related conferences, workshops and seminars, but also the big conferences in virology and bioinformatics. The listed events are covered in the newsletter to inform people about approaching deadlines. EVBC members involved in those events are highlighted. Subscription to the calendar is free also to non-members. All upcoming lectures and workshops of the EVBC (see Section 2.3 and Section 2.4) are announced in the newsletter. EVBC members can also send us job vacancies to reach a large audience of interested candidates. Since news on SARS-CoV-2 took a large share in our monthly newsletter, we have been publishing a monthly Special Issue Newsletter on SARS-CoV-2 since the beginning of the pandemic (in parallel to our regular newsletter).

The EVBC newsletter is publicly available to all interested readers. The EVBC is also maintaining a Twitter channel to spread recent publications, vacancies and event announcements.

### 2.2. Meet the Experts: The International Virus Bioinformatics Meeting

Bringing together bioinformaticians and virologists is crucial to discuss fundamental computational issues arising from specific virological questions. The annual conference of the EVBC offers the opportunity to exchange the latest research results and experiences with an extensive network of leading experts and aspiring young scientists in both areas and engage in inspiring discussions. The conference has a clear focus on bioinformatics approaches in virology, reaching from systems virology; virus-host interactions; immunology; viromics and metagenomics; virus identification, classification and annotation; virus evolution; viral ecology to epidemiology and surveillance. For people newly entering the field of virus bioinformatics, this meeting is a focal point to gain an insight into the state-of-the-art of the research landscape.

The annual conference of the EVBC started off with the founding meeting in 2017. Over the years, the meeting developed from an annual get-together of the EVBC to an international conference far beyond the members of the EVBC. In 2019, a selection process for contributed submissions was included to open-up the scientific program [5] and the number of high-quality submissions is increasing every year. In 2020, the pandemic has made us even more aware of the importance of accelerating research in viral bioinformatics. The conference was originally planned to take place in Bern, Switzerland, but had to be postponed and converted to an online format. Despite concerns being raised about the lack of opportunities for face-to-face discussions in virtual meetings, the participants created a highly interactive scientific environment [6]. In addition, we experienced an increase in registrations after announcing the online meeting. In 2022, we continued the success of the virtual format with an incredibly high amount of registered participants (see Table 2).

Each year, the conference is jointly organized by the EVBC and a local organizing team of EVBC members (see Table 2). The organizing committee sets high standards for the quality of the scientific program. Invited lectures by renowned scientists whose work has inspired researchers in virology and bioinformatics are complemented by talks on recent advances by experts at all career stages, selected from the contributed submissions. The conference is organized without registrations fees and with a very personal atmosphere. The conference is attracting scientists far beyond the EVBC community and a lot of researchers are returning participants. For some of our members, the conference even became their favorite conference of the year.

During the conference, the EVBC members are getting together for the EVBC general annual meeting. During this meeting the EVBC office is presenting new and ongoing services and the achievements of the EVBC of the past year. The members are discussing future directions of the EVBC. Also, every three years, the Board of Directors is elected by the members during this meeting.

### 2.3. Dive into Virus Bioinformatics: The EVBC Lecture Series

One aim of the EVBC is to intensify and further improve the qualification of graduates of virology, bioinformatics and related topics, in the field of virus bioinformatics. To train early career researchers and to keep experts at all career stages up to date with the latest developments, the EVBC is organizing two monthly lecture series. Both lecture series are organized in a virtual format and participation is free of charge.

The “viruses in silico” lectures series is highlighting new approaches and tools in virus bioinformatics. The lectures were initiated after the success of the virtual format of the annual conference in 2020. Some of the past speakers, featured below, include Caroline C. Friedel (see Section 3.4) speaking about lessons for virus bioinformatics learned from her research on HSV-1 infection, Denise Kühnert (see Section 3.2) speaking about phylodynamic modeling of modern and ancient infectious disease dynamics, and Katy Brown (see Section 3.1) speaking about CIAlign, a tool to clean, interpret and visualize multiple sequence alignments, and its application to virus discovery. The lectures are attended by 20–80 participants each month.

The ECR Viromics Webinar Series is aimed at early career researchers studying viruses in complex communities. Here, we open up a platform for ECRs to present and discuss their research and get connected. The lecture is jointly hosted by the Center of Microbiome Science at Ohio State University, the EVBC, and the NSF EMERGE Biology Integration Institute.

### 2.4. Learn the Basics: Virus Bioinformatics Workshops

Next to learn about current advancements during lectures and conferences, the EVBC aims at laying the foundation for a high quality education in virus bioinformatics. The EVBC is organizing online and on-site workshops and training courses covering theoretical and practical aspects in virology, bioinformatics, data science, etc.

EVBC is also offering support in organizing bigger on-site workshop events, such as the International Bioinformatics Workshop on Virus Evolution and Molecular Epidemiology (VEME) organized by Anne-Mieke Vandamme (see Section 3.5).

### 2.5. Find What You Need: The Virus Bioinformatics Tools Collection

Recent biotechnological developments have led us into the age of big data and the virus bioinformatics community is constantly developing newly required sophisticated tools for extracting and interpreting relevant information from virus data. The EVBC is curating a collection of these tools. We use several properties to describe each tool: tags for the category of application (e.g., phylogenetics, annotation, epidemiology), tags regarding software development (e.g., software type, coding language, repository), the license, tool website and respective publication, the virus families for which the tool was either designed for or validated on, the EVBC members involved. All properties are part of the search and filter function. In addition, the collection contains several presorted views to navigate through the tools, e.g., sorted by category, virus family or software type.

Note that the EVBC is not maintaining the tools in the collection. However, in the future, we would like to organize assessment strategies to point researchers to the state-of-the-art tools for their analysis and point out advantages and drawbacks of the different methods.

During the pandemic, bioinformatics tools designed explicitly for SARS-CoV-2 have been developed as a rapid reaction to the need for fast detection, understanding and treatment of COVID-19. The EVBC was pointing out relevant methods to combat COVID-19 and accelerate SARS-CoV-2 and coronavirus research [4].

### 2.6. Achieve More Together: Collaborative, Jointly Funded Projects

In addition to bringing together scientists during conferences, workshops and seminars to promote scientific exchange, the EVBC is initiating large collaborative research projects. The EU funded Marie Sklodowska-Curie Innovative Training Network VIROINF [9] focuses on (harmful) virus-host interactions by combining virus research with specifically designed bioinformatical tools to avoid infections, and enables vaccinations and treatments. The consortium consists of 28 high-profile principal investigators, 22 of whom are EVBC members. Some of the female PIs in this network are introduced below (see Section 3.2, Section 3.3 and Section 3.4).

The EVBC also aims to strengthen the exchange of ideas and initiate scientific cooperation between industrial members and scientists. Both sectors have a significant interest in viral bioinformatics but often approach the subjects from different perspectives and have different requirements for the development and use of research tools. The VIROINF network is providing young scientists with training opportunities in non-academic research environments to leave them better equipped to collaborate with non-academic partners or follow non-academic research careers.

### 2.7. Get Published: Special Issues on Virus Bioinformatics

Scientific results of an interdisciplinary research field are often difficult to publish in journals of either the one or the other area. Therefore, the EVBC is running a recurring special issue on virus bioinformatics since 2018. In addition, we promote special issues that are edited by our members.

The first special issue on Virus Bioinformatics was published in Virus Research in 2018 covering 12 research articles and short communications. Since 2019 we are running the special issue in MDPI Viruses, covering 18 (in 2019) and 15 (in 2020) papers. The recent issue “Virus Bioinformatics 2022” is still under review and is so far covering 12 papers.

## 3. Women in the EVBC

The EVBC seeks to provide a supportive environment for talented researchers that is free of gender bias, for example by ensuring adequate participation of female speakers in seminars, courses and conferences. Since EVBC was founded, we have succeeded in steadily increasing the proportion of female members. However, of the 247 members of the EVBC, still only about 28% are women scientists (see Figure 1). The gender gap in science persists to a greater degree than in other professions, but in particular in high-end, math-intensive fields such as computer science and engineering [10,11,12,13].

The more women get involved in science – or any other traditionally male-dominated field—the greater the overall knowledge in that field becomes [14,15]. Collaboration is now the foundation in most research areas but in particular in interdisciplinary fields. The inclusion of qualified women enriches the creativity, flexibility and insights of research projects and increases the chance of innovation; moreover, it is an competitive advantage.

We should be aware that this is not only affecting female scientists but our society in general. What science decides to solve, for whom things are developed, and how science is conducted is strongly related to who carries out the scientific research. As an example, for years, animals used in scientific experiments were predominantly male so as not to falsify experimental results with female hormone fluctuations [16,17,18]. It was simply assumed that male animals could be used to reliably predict the effects on both, men and women. The variable “sex” was not systematically considered like other variables. We need to prevent science from being conducted only from a merely male perspective. Women are not only the other sex that must also be considered or simply the deviation from the male norm. Women are about half of society. The more women in research, the broader the spectrum of research topics and viewpoints, and consequently the broader the range of inventions and breakthroughs that result from a different, non-male perspective. These aspects can be extended to other types of diversity [19,20], from ethnicity and nationality, to scientific discipline and professional experience. Although diverse teams are harder to manage than homogeneous groups, they perform better in innovation, problem-solving, flexibility, and decision-making [14].

To keep more talented women in research, we need to ensure that they are not kept at lower levels but are given the opportunity to lead the field [21,22]. Often, women have to be much better qualified to occupy the same positions as men. On the other hand, women have fewer role models to inspire their interest in science. Thus, we would like to take this opportunity to highlight the achievements of several outstanding female scientists in the EVBC.


**Dr. Franziska Hufsky: Coordinating EVBC activities.**


Franziska Hufsky studied bioinformatics at the Friedrich Schiller University Jena, Germany, where she also got her doctorate degree developing novel methods for the analysis of small molecule fragmentation mass spectra. She then joined the group of Manja Marz (see Section 3.4) helping to build up the EVBC.

Franziska has been a founding member of the EVBC. She is compiling the monthly newsletter, monitoring the publications of EVBC members, filling the EVBC Twitter channel, maintaining the EVBC homepage including the calendar and vacancies services, and curating the collection of virus bioinformatics tools. She was in the organizing committee of all annual conferences (except 2018) and is co-editor of all special issues on virus bioinformatics in MDPI Viruses. Franziska is organizing the viruses in silico lecture series, partly organizing the ECR Viromics Webinar Series and taking care of the EVBC workshop program. She is organizing joint publications by EVBC members [4,23,24], including the conference reports [5,6,8]. She coordinated the grant application for VIROINF. In addition, she is involved in several (mainly German) outreach activities on virus bioinformatics [25,26,27,28] (see http://bioinfowelten.uni-jena.de/ and http://www.formeltiere.de/ (accessed on 1 July 2022)) and is in the equal opportunities team of the Friedrich Schiller University Jena.

### 3.1. Virus Discovery


**Prof. Dr. Lia van der Hoek: Anelloviruses, coronaviruses, and novel viruses.**


Lia van der Hoek is a virologist at the Amsterdam University Medical Center, location AMC, University of Amsterdam. She developed a novel virus discovery method, named VIDISCA, that can detect any RNA or DNA virus. Using this technique she was the first to discover human coronavirus NL63, in 2003, an endemic human coronavirus that remained unrecognized till then [29]. It soon became clear that HCoV-NL63 had spread worldwide with continuous reinfections throughout life, and the average age of first infection being around the first years of life [30]. Besides the many research papers on HCoV-NL63 and the other endemic human coronaviruses, Lia continued with virus discovery and her group found several other previously unknown human and animal viruses which she could link to unexplained diseases [31,32]. The latest item under investigation by Lia is the “anellome”, the total of ubiquitously found chronically present anelloviruses in humans of which it is still unknown whether they are involved in health or disease [33].

Lia has been a founding member of the EVBC and was an invited speaker at the founding meeting of the EVBC introducing the HONOURs training network [34] to the community.


**Dr. Katy Brown: Virus discovery, genomics and evolution.**


Katy Brown is a postdoctoral researcher and bioinformatician at the University of Cambridge, working in Andrew Firth’s group. Her research is focused on characterizing new and poorly understood viruses. Her work involves screening high throughput sequencing data to identify RNA viruses, with a current focus on viruses of bees and associated arthropods. She has also worked on a number of projects on non-canonical translation in viruses [35,36]. She also develops bioinformatics tools, such as CIAlign [37], a multiple sequence alignment cleaning and visualization tool which was co-developed with Charlotte Tumescheit (see Section 3.4). Katy is involved in promoting good software development practices as a fellow of the Software Sustainability Institute.

She completed her PhD at the University of Nottingham in 2014, with a focus on identifying and characterizing ancient viral elements in the DNA of modern primates and rodents [38,39]. She went on to complete the MRC Computational Genomics Analysis and Training (CGAT) fellowship program [40] at the University of Oxford, collaborating on a series of projects focused on human disease [41]. She also has a BSc. in Biology and an MSc. in Quantitative Genetics and Genome Analysis.

Katy joined the EVBC in 03/2019 and regularly attends the annual conference, where she also presented her work in 2019. She has given a lecture in the viruses in silico lecture series and her software CIAlign [37] is part of the EVBC tool collection.


**Dr. Sofia Paraskevopoulou: Discovering new viruses in silico.**


Sofia Paraskevopoulou is a biologist by training who, after completing her major’s studies at the University of Crete in Greece, moved on to pursue a Masters in evolutionary biology at the University of Bonn in Germany. It was during her Masters when she discovered the world of bioinformatics and received training in writing code. She obtained a Dr. rer. nat. degree for the work that she did on virus evolution and computational virus discovery at the Charité Institute of Virology in Berlin, under the supervision of Christian Drosten.

She was involved in the discovery of hundreds of novel RNA virus genomes in insects [42,43], many of which have established new virus families in the taxonomy of viruses [44,45]. The phylogenetic placement of newly discovered viruses in the context of already known ones, particularly when combined with information on virus genome organization and virus host species, offers insights into the general understanding of RNA virus evolution. She has also participated in the discovery of the first non-human mammalian deltavirus [46] which resulted in the establishment of *Ribozyviria* [47], a new realm of satellite nucleic acids, with implications on the field of viroid research.

Sofia is now a postdoctoral researcher at the Robert Koch Institute in Berlin where she has joined forces in the genomic molecular surveillance of SARS-CoV-2. Providing outbreak analytics, detecting recombinant genomes, and monitoring variants and mutations of concern are parts of her daily work.

She joined the EVBC in 10/2020 and presented her work at the annual conference in 2020 [6]. She finds the EVBC community a welcoming place that values its members and audience, and she is regularly attending the lectures, workshops, and conferences organized by the EVBC.


**Luca Nishimura: Discovering ancient viruses.**


Luca Nishimura graduated from Hokkaido University in Japan. When she was an undergraduate student, she participated in the international Genetically Engineered Machine (iGEM) contest and proceeded with experiments on synthetic biology. After graduation, she changed her research field to virology. Currently, she is a PhD student at the Graduate University for Advanced Studies (SOKENDAI) in Japan and working on ancient viral studies.

Luca’s research aim is to discover the ancient viruses more comprehensively and elucidate long-term viral evolution. To investigate ancient viruses, she has utilized whole genomic sequencing (WGS) data derived from archaeological samples, such as teeth, bones, and coprolites. She has conducted bioinformatic analyses to handle a vast amount of WGS data and tried to detect ancient viruses and comprehend the ancient viromes. Moreover, she has investigated the viral evolution using those ancient viral genomes.

Luca joined the EVBC in 03/2022. She gave a poster presentation at annual conference in 2022 and won the Best Poster Award [8]. Also, her member profile was featured in the EVBC Newsletter Vol. 60 (05/2022).

### 3.2. Virus Evolution


**Dr. Denise Kühnert: Phylodynamics of modern and ancient pathogens.**


A mathematician by training, Denise Kühnert has a highly interdisciplinary background. She dove into phylogenetics in 2009 and obtained her Ph.D. in Computer Science from the University of Auckland in 2014. Thereafter, she held two consecutive postdoctoral positions in integrative biology at the ETH Zürich and in infectious diseases at the University Hospital Zurich, Switzerland. In 2018, she started her independent Max Planck Research Group at the Max Planck Institute for the Science of Human History.

During her Ph.D., Denise was among the first to combine phylogenetics with mathematical epidemiology in order to use virus genomes to understand viral transmission dynamics [48,49,50]. Fascinated by the insights that can be gained from pathogen genomes, Denise continued developing phylodynamic methods for the analysis of rapidly evolving pathogens as a postdoctoral researcher [51,52]. When starting her own research group, Denise expanded her research focus to ancient pathogens. While continuing to track the spread of modern pathogens like SARS-CoV-2 [53] she shed light on virus evolution over millennia, such as Hepatitis B virus [54].

Denise joined the EVBC in 10/2017 and recently presented her research in the viruses in silico lecture series. She taught virus phylodynamics at the 23rd VEME conference in Berlin, co-organized by EVBC, and is a co-author of the overview of computational tools for SARS-CoV-2 research [4].


**Prof. Dr. Ana Barroso Abecasis: Phylodynamics and antiviral drug resistance.**


Ana Barroso Abecasis is Assistant Professor at Institute for Hygiene and Tropical Medicine (IHMT), NOVA University of Lisbon. She trained as Medical Doctor (NOVA University of Lisbon, 2014), has a previous degree in Pharmaceutical Sciences and specialization degrees in Bioinformatics and Microbiology. She did her PhD in Medical Sciences at the KU Leuven (Belgium, 2009) working in evolutionary dynamics of HIV-1.

She coordinates the research line of Global Pathogen Dispersion & Mobility of Populations at the Global Health and Tropical Medicine Research Center. She is member of several European networks of phylodynamics and antiviral drug resistance, including European Virus Bioinformatics Center, EuResist and European Society for Translational Antiviral Research. Her research interests include virology, antiviral drug resistance, bioinformatics, evolution and global health.

Ana has published around 81 articles in international peer-reviewed journals, with an h-index of 26 (Google Scholar) and 19 (ResearcherID). In 2012, received the L’Oreal Portugal FCT Unesco Medal of Honour for Women in Science. She teaches at several Masters and Doctoral Programmes at the IHMT and NOVA University.

Ana has been a founding member of the EVBC. She has organized and taught the Workshop on Virus Evolution and Molecular Epidemiology 2018 (VEME) with Anne-Mieke Vandamme (see Section 3.5) together with the EVBC. She is co-author of VIRULIGN [55] and ArboTyping [56] in the virus tool collection. Ana is regularly reading the newsletter to stay updated.


**Dr. Urmila Kulkarni-Kale: Virus evolution, diversification and typing.**


Urmila Kulkarni-Kale is a computational virologist and senior scientist at the Bioinformatics Centre, Savitribai Phule Pune University, India. With 30+ years of experience, she has made several contributions to the fields of virus bioinformatics, immunoinformatics and phyloinformatics. Urmila is elected fellow of the Maharashtra Academy of Sciences and a member of the editorial board of the Elsevier journal ‘Infection Genetics & Evolution’.

Intrigued by the diversity of viruses, Urmila has been analysing a variety of virus data to answer some of the most fundamental questions on evolution, diversification, spill overs and emergence of immune escape for viruses. Urmila has successfully built a multi-disciplinary research program and successfully applied population genetics concepts in virology to explain lineage emergence and diversification of RNA viruses. Her group has developed novel algorithms for epitope predictions [57], alignment-free phylogeny and designed servers for virus tying [58,59,60]. She has active collaborations with national and international virologists and has worked towards investigating virus outbreaks and pandemics including COVID-19 [61,62,63]. With a focus on translation, Urmila has contributed to computational vaccine design (JEV, HPV16), to explain the escape of viruses from vaccine-induced neutralization (Mumps) and to understand potential impact of virus-host variations on efficacy of vaccine (COVID-19).

Urmila joined the EVBC in 08/2020. She is co-author of several virus typing servers [58,59,60] in the tools collection and is regularly attending the annual conference.


**Dr. Magda Bletsa: Characterizing viral evolutionary processes.**


Magda Bletsa is a post-doctoral researcher at the Department of Hygiene, Epidemiology and Medical Statistics of the National and Kapodistrian University of Athens (Greece) and an external researcher at the Department of Microbiology, Immunology and Transplantation at the Rega Institute (Belgium). She holds a Bachelor’s degree on Biochemistry and Biotechnology and a Master’s degree on Molecular Biology and Genetics. In January 2021, she received her PhD from KU Leuven (Belgium) for her work on the molecular epidemiology of HIV and hepaciviruses [64,65], followed by a first post-doc in the the group of Philippe Lemey.

Her research aims at a better understanding of viral ecology and evolution. She is particularly interested in characterizing the evolutionary processes and dynamics underlying pathogen genomic diversity across multiple scales. This goes from studying within- to between-hosts evolution, reconstructing deep and recent evolutionary events up to surveying both the virus and their hosts’ genomes [66,67,68]. To this aim, she uses a combination of wet-lab approaches (molecular biology methods, high-throughput sequencing) and computational tools (bioinformatics, phylogenetics, phylodynamics) [69].

Magda joined the EVBC in 11/2021. She is actively attending the monthly lecture series and the annual conference. At ViBioM 2022 [8], she chaired the session on “Viral emergence and surveillance”.


**Prof. Dr. Karin Metzner: HIV genomics and viral escape.**


Karin Metzner is a molecular virologist and research group leader at the Department of Infectious Diseases and Hospital Epidemiology, University Hospital Zurich, Switzerland. She has a broad and long-standing experience in the research field of viral genetics and molecular biology including next-generation sequencing of complex virus populations [70,71]. Her research interests includes but are not limited to the evolution of HIV-1 [72] and the mechanisms of viral escape from the selective pressure of antiretroviral drugs, viral and cellular factors underlying HIV-1 latency including HIV-1 integration sites, and the development of gene therapy approaches to combat HIV-1.

Karin joined the EVBC in 02/2021. She is part of the VIROINF network [9]. The EVBC newsletters and special issues on SARS-CoV-2 keep her up to date on these dynamic areas of research. The tool collection is very valuable. She is co-author of V-Pipe [73] in the virus toll collection.


**Dr. Maryam Zaheri: Clinical metagenomics and HIV vaccine research.**


Maryam Zaheri is a scientist and bioinformatician in the Institute of Medical Virology of University of Zurich where she is involved in computational virus research with the main focus on metagenomics analysis of clinical samples [74,75], HIV vaccine discovery and bioinformatics analysis of SARS-CoV-2 (since the beginning of the pandemic) [76,77].

Maryam obtained her B.Sc. from University of North Carolina in Charlotte with the final project on differential gene expression analysis [78] and MSc from École polytechnique fédérale de Lausanne defending her thesis on gene regulatory network inference [79] under supervision of Bernard Moret. In 2014, she received her PhD in computational evolutionary biology with the focus on developing a mathematical framework to model codon evolution [80,81] from University of Lausanne in the group of Nicolas Salamin.

Since then and until 2018 when she started her current role, she worked as a researcher at ETHZ (the group of Niko Beerenwinkel) with the focus on viral next generation sequencing analysis, as well as a bioinformatician at Roche and Novartis.

Maryam joined the EVBC in 08/2019. She has attended several EVBC lectures, as well as the annual conferences since 2019. She reads the newsletters to stay updated.

### 3.3. Viral Infection and Immune Response


**Dr. Stefanie Deinhardt-Emmer: Elucidating pathomechanisms.**


As physician, Stefanie Deinhardt-Emmer entered the field of virology during her medical thesis (PMID: 27426251). After completing clinical experience, she began her specialization in microbiology, virology, and infectious diseases. She received her first fellowships in Clinician Scientist programs, giving her the opportunity to focus on experimental research and having a research fellowship in CA, USA [82]. She was fortunate to have two strong women to guide her in her clinical and scientific interests.

Currently, Stefanie is an assistant medical director, leading a group on translational microbiology/virology at Jena University Hospital, Germany, and coordinating a BMBF collaborative project on SARS-CoV-2. Particularly important to her is the flexibility to develop new projects and the opportunity to work anywhere—at home with her children, at congresses, in the lab or in the office—to achieve her research vision: the elucidation of pathomechanisms between viruses, bacteria and the host [83,84,85,86,87,88,89,90,91]. Therefore, bioinformatics is a great opportunity and a great inspiration to reach her goals.

Stefanie has been a founding member of the EVBC. She was an invited speaker at the annual conference in 2019 in Glasgow [5].


**Dr. Claudia Claus: Prenatal virus infections.**


Claudia Claus is a senior researcher at the Institute of Medical Microbiology and Virology of the Medical Faculty of Leipzig University, Germany. She studied Biological Sciences at Martin-Luther-University Halle-Wittenberg, Germany. During her enrollment at Queen’s University Belfast, UK as a non-graduating student she attended courses with a special focus on virology. Her research stay as a visiting scientist at Georgia State University, GA, USA during her Ph.D. studies at Leipzig University deepened her interest in the study of prenatal virus infections with a focus on rubella virus. During her postdoctoral research she used an induced pluripotent stem cell-derived cell culture model to study rubella virus infection-induced alterations during the very early steps of human development [92]. She obtained a postdoctoral lecture qualification (habilitation) in experimental virology. Currently her research focus includes virus infection-associated metabolic alterations of the RNA viruses rubella virus and Usutu virus and their implication in the associated innate immune response [93,94].

Claudia joined the EVBC in 08/2019. She is attending the annual conference as well as EVBC lectures.


**Lore Van Espen: Human gut virome in chronic liver disease.**


Lore Van Espen is a PhD researcher at the KU Leuven, Belgium, where she also obtained her BSc and MSc in Biomedical Sciences in respectively 2016 and 2018. She currently works at the Laboratory of Viral Metagenomics, Rega Institute, and is supported by an FWO fellowship. Her research focuses on the human gut virome, including eukaryotic viruses and phages, in patients with chronic liver disease. The gut viromes are studied using the in-house developed NetoVIR protocol [95] and a combination of optimized bioinformatic tools. A first aim of the project was to develop a viral genome catalog (DEVoC), that led to the identification of a previously undescribed prevalent phage [96]. Currently, her major focus point is to study the interaction between phages and their bacterial hosts. Her PhD project is part of two European consortia, MicrobLiver and MICROB-PREDICT (https://microb-predict.eu/), both aiming at understanding the role of the global gut microbiome in chronic liver disease by integrating host and microbial multi-omics to identify diagnostic and/or predictive biomarkers.

Lore joined the EVBC in 04/2021 and has attended several EVBC lectures, as well as the annual conference since 2018.


**Prof. Dr. Li Deng: Functional role of human virome.**


Li Deng studied Environmental Engineering at the Tsinghua University, China (BEng), Environmental Science at the University of Nottingham, UK (MSc), and Microbiology at the University of Bristol, UK (PhD). She studied freshwater cyanophages during her PhD under the supervision of Paul Hayes. She received post-doctoral training at the University of Arizona, US (supervisor: Matthew Sullivan) where she developed interests in using omics tools to address interactions between viruses and hosts, especially in a cultivation independent and single-cell manner. At the Helmholtz Centre Munich, Germany, she was head of a DFG Emmy Noether junior group, as well as Helmholtz Young Investigator group. She was promoted to W3 Professor of “Microbial Disease of Prevention” at the Technical University of Munich in 2022.

Li conducts research in the area of phage biology. In contrast to the classical approaches, the Deng lab uses single-cell technologies, multi-omics, culture-independent techniques, and machine learning to isolate novel phages, develop phage-based therapeutics, and understand the underlying mechanisms of phage-host interactions and their impact on the human host [97] They apply the knowledge gained for targeting pathogenic bacteria and ameliorating disease severity by restoring healthy microbiota in multiple diseases or conditions [98,99,100,101].

To facilitate the acceptance of phage therapy, which uses phages to target pathogenic bacteria the Deng lab has developed highly efficient phage cocktails against multiple critical multiresistant bacteria. In addition, they promote increasing collaborations between scientists and physicians toward defining a roadmap for future translational phage research in Germany [102,103].

Li is a founding member of the EVBC and on the board of directors since then. She gave an invited talk at the founding conference in 2017. Li is also a principal investigator in the VIROINF network [9].


**Dr. Patricia Agudelo-Romero: Host-virus interaction phatosystem.**


Patricia Agudelo-Romero is a computational biologist interested in the integration of anti-viral responses of the host using different omics, such as transcriptomics and metabolomics. She holds an undergraduate engineering degree from the University of Llanos (Colombia) along with a Masters and PhD degree from The National School of Public Health of Spain and the Polytechnical University of Valencia. Since, she has held postdoctoral positions in the University of Lisbon (Portugal), as well as the University of Western Australia and the Telethon Kids Institute (Australia). Through her postdocs, she has demonstrated great agility transitioning from plant to medical science and established a research platform.

Currently, at Telethon, she is leading the bioinformatics research in the Respiratory Health for Kids team, investigating host-virus interactions in chronic respiratory diseases. She has developed bioinformatics pipelines for omics integration, viral dual RNA sequencing and viral contigs assembly and characterization (https://agudeloromero.github.io/EVEREST/ (accessed on 5 July 2022)). Her main purpose is to identify antiviral signatures that can be used as biomarkers to work out new treatment options in children with chronic respiratory diseases and exploring its translation into standard care.

Patricia joined the EVBC in 11/2021 and is excited about the regular EVBC events.


**Prof. Dr. María Inés Gismondi: Studying virus-host interactions.**


María Inés Gismondi obtained her PhD in Virology working on HCV evolution during chronic infection in children at the Ricardo Gutiérrez Children’s Hospital in Buenos Aires, Argentina [104,105,106,107,108,109,110]. In 2008, she moved to the field of veterinary virology at the Institute of Agrobiotechnology and Molecular Biology (INTA-CONICET) [111,112,113,114]. Currently, her research is focused on studying virus-host interactions in FMDV-infected cells and animal models. She is also a Professor of Virology and Cellular and Molecular Biology at the University of Luján, Argentina, where she enjoys motivating students with innovative educational strategies.

From the beginning of her career, María has applied bioinformatics to study virus evolution, predict virus genomic structures or characterize viral populations from genomic data [115]. In recent years, her group also developed some bioinformatics tools for the analysis of viral infection and virus-derived sequence data.

María joined the EVBC in 08/2019 and has participated in the annual conferences, where she enjoys getting in touch with bioinformaticians from all over the world to learn from them how to think in a ‘computational’ manner. In December 2021, she organized the virtual activity “Getting involved in the development of a new virus database”, led by Manja Marz (see Section 3.4), to increase EVBC visibility in Argentina. She is co-author of ViralPlaque [116] and Covidex [117] in the virus tool collection.

### 3.4. Virus-Host Transcriptomics


**Prof. Dr. Caroline C. Friedel: Virus-host transcriptomics.**


Caroline C. Friedel is a bioinformatician by training, having obtained her B.Sc. and M.Sc. degrees in the newly minted joint bioinformatics program of the Ludwig-Maximilians-Universität (LMU) and Technical University (TU) Munich in 2003 and 2005, respectively. She obtained her Ph.D. in bioinformatics from the LMU in 2009 under the supervision of Ralf Zimmer and, after a short postdoctoral position, moved to Heidelberg University for an assistant professorship in 2010. She later returned to LMU and obtained a tenured associate professorship at the LMU in 2015.

During her Ph.D., Caroline already investigated virus-host interactions, initially with a focus on protein-protein interactions for SARS [118] and herpesviruses [119]. With the advent of high-throughput sequencing technologies, which allow sequencing both host and viral RNA in one experiment, analysis of host transcriptomic responses to virus infection and modulation of host transcription by viruses has become a major part of her research. While this initially focused on the many ways that Herpes simplex virus 1 impacts host transcription [120,121,122,123,124], she is now extending her research to other DNA viruses in the newly funded DFG research unit DEEP-DV.

Caroline is a founding member of the EVBC and recently presented her research in the viruses in silico lecture series. She is co-author of the HSV-1 Viewer [125] in the virus tool collection. She is also a principal investigator in the VIROINF network [9].


**Prof. Dr. Irmtraud M. Meyer: Detecting functional RNA structures and RNA-RNA interactions.**


Irmtraud M. Meyer obtained a M.Sc. in Physics from the RWTH Aachen, Germany. She then joined Richard Durbin’s Bioinformatics research group at the Sanger Institute in Cambridge, UK, as a Wellcome Trust Prize Student and External Research Scholar of Trinity College, obtaining her PhD in 2003. After three years as a post-doc at the University of Oxford and the European Bioinformatics Institute in Cambridge, UK, she became a group leader and faculty member at the Centre for High-Throughput Biology and the Department of Computer Science at the University of British Columbia in Vancouver, Canada. Since 2016, she is a Senior Group Leader at the Berlin Institute for Medical Systems Biology at the Max Delbrück Center and Full Professor at the Freie Universität in Berlin, Germany.

Irmtraud is keen to investigate how the expression of genes is regulated within the transcriptome in vivo, from eukaryotes to various viruses. To that end, her research group has contributed a range of unique computational methods and analyses that are capable of detecting subtle signals within transcripts which regulate their expression in vivo [126,127,128,129,130,131,132,133,134,135,136,137]. One particular focus of her computational research group are processes that regulate the splicing of transcripts as well as processes that are mediated by RNA structure features or *trans* RNA-RNA interactions between different transcripts.

Irmtraud joined the EVBC in 10/2020, contributing an invited lecture to the annual conference in 2020 in Basel [6] and participating in the regular EVBC events.


**Prof. Dr. Manja Marz: Bioinformatical tools for sequence analysis of RNA viruses and their hosts.**


Manja Marz obtained a diploma in Biology and Computer Science from the Leipzig University, Germany. After her PhD in Bioinformatics, she was immediately appointed to a group leader position at the University of Marburg, Germany, and after only two years she was offered a junior professorship at Friedrich Schiller University Jena, Germany. She became the youngest female professor to receive her full professorship three years later. During her time in Jena, she has helped shape the university by founding the Aging Research Center Jena, the Michael Stifel Center Jena for data driven Science and the European Virus Bioinformatics Center in 2017.

Manja tries to bridge the world of virology and bioinformatics by (i) developing tools (especially for RNA viruses, as Manja has ever since focused on non-coding RNAs), (ii) applying existing tools, and finally (iii) trying to work theoretically on the existence and meaning of viruses.

Together with her students, Manja developed tools to predict conserved long-range RNA-RNA interactions in full viral genomes [138], which has been applied to several viruses such as HCV [139]; to analyze long-range RNA-RNA interactions in viral genomes with structured RNA [140]; to predict linear B-cell epitopes based on neural networks [141]; and to generally predict viral hosts [142]. Manja studied coronaviruses prior to the 2019 outbreak in terms of RNA genome secondary structure analysis [143,144,145], including a study to investigate genome modifications [146]. She attempted to study the virome in groundwater by developing new protocols in the wet lab [147] and combining multiple techniques and developing software for analysis [148]. She also applied and extended existing tools when she studied differential transcriptional responses to Ebola and Marburg virus infections in bat and human cells [149] or analyzed virus- and interferon-alpha-induced transcriptomes of cells from microbats [150]. Manja has authored several summary publications on viral software [4,151], the needs in the field [23,24,152], the annual meeting of the EVBC [5,6,7] or a collection of viral genomes based on alignments in the Rfam database [153].

Manja founded the EVBC in 2017 and has been EVBC director since then. Together with Franziska Hufsky, she is adding new features to the EVBC, from the annual meeting, to a collection and testing tools, a newsletter, etc., to the creation of a viral database and the organization of workshops upon user request.


**Dr. Charlotte Tumescheit: RNA structure prediction in RNA viruses, Protein Structure, MSAs.**


Charlotte Tumescheit is a postdoctoral researcher in Martin Steinegger’s group at Seoul National University in Seoul, Korea, where she is currently part of the team developing Foldseek [154], a tool for fast protein structure search in large datasets, as well as aligning multiple protein structure sequences.

While originally trained in mathematics, Charlotte became interested in bioinformatics and its applications to viruses and developed a tool to predict RNA secondary structure in RNA viruses during her PhD studies at the University of Cambridge under the supervision of Andrew E Firth. Here, she also developed CIAlign [37]—together with Katy Brown (see Section 3.1). CIAlign cleans, interprets, and visualises multiple sequence alignments. She also performed bioinformatics analyses related to virus discovery, curation of multiple sequence alignment of viruses, and protein foldability [35,155,156,157].

Charlotte joined the EVBC in 10/2019 and CIAlign is part of the EVBC tool collection. She attended the annual International Virus Bioinformatics Meeting in 2020 [6], where her CIAlign poster won the best Scientific Poster Award.

### 3.5. Emerging Viruses and Epidemiology


**Prof. Dr. Anne-Mieke Vandamme: Molecular epidemiology of viruses.**


Anne-Mieke Vandamme was trained as biochemist, and holds a PhD in sciences. She joined the Rega Institute at the University of Leuven in 1990, where she started a unit on virus genetic testing for clinical and epidemiological purposes. She co-founded a new division, Clinical and Epidemiological Virology; and co-started the “Institute of the future”, a transdisciplinary research incubator. She is member of the Superior Health Council of the Belgian government and of the WHO ad-hoc advisory group on COVID-19.

Anne-Mieke and her team perform research on the molecular epidemiology of viruses, on virus drug resistance testing in clinical context, and on bioinformatics (data mining, phylogenetic analysis) tools. She supported the development of a few widely used (bioinformatics) tools: the Rega HIV Drug Resistance algorithm, the Rega HIV typing and subtyping tool, RegaDB, and Genome Detective [158]. She is currently coaching a Coronavirus Pandemic Preparedness challenge and is coordinating a project on Futures for Food. She authored more than 400 scientific publications in international journals and more than 800 presentations for International Meetings.

In addition to her research activities, Anne-Mieke is founder and organizer of the yearly International Bioinformatics Workshop on Virus Evolution and Molecular Epidemiology (VEME, 26th in 2022), co-initiator and co-organizer of the European HIV Drug Resistance Workshop (20th in 2022), invited professor at the Universidade Nova de Lisboa, Portugal and visiting professor at the University of Belgrade, Serbia.

Anne-Mieke has been a founding member of the EVBC and organized the 23rd VEME together with the EVBC. She is co-author of GenomeDetective [158] and ArboTyping [56] in the tools collection.


**Prof. Dr. Emma Thomson: Understanding new and emerging viral infections.**


Emma Thomson’s lab focuses on the use of next generation sequencing and functional assays to investigate viruses that present a risk to human health in the UK and Uganda. Emma’s work focuses on understanding how viruses evolve under immune pressure and with the use of antiviral treatments. Working with the Clinical Trials Facility in the Queen Elizabeth University Hospital, Emma is the principal investigator for NHS GG&C for several vaccine trials, including the phase III Oxford/AZ ChAdOx1 vaccine, the Novavax phase III trial, the Valneva phase III trial and the COV-BOOST and DOVE studies.

In March 2020, her lab sequenced the first case of SARS-CoV-2 in Scotland and continued to carry out high-scale sequencing in order to monitor introductions of virus variants in the country. Working with PHS, PHE and other partners, Emma’s lab is a member of the Genotype2Phenotype (G2P) consortium, the UK Coronavirus Immunology Consortium (CIC) and ISARIC4C with the role to investigate how changes in the genome of the virus may translate to changes in the behaviour. Additionally, she has also investigated several emerging viruses that have been detected in recent years in the Scottish population, including Ebola virus and Seoul hantavirus.

Emma joined the EVBC in 02/2022. She is co-author of GLUE [159] in the virus tool collection.


**Dr. Evangelia Georgia Kostaki: Applying molecular epidemiology to public health.**


Evangelia Georgia Kostaki is a post-doctoral researcher at the Department of Hygiene, Epidemiology and Medical Statistics of the National and Kapodistrian University of Athens (NKUA) Medical School in Greece. She holds a bachelor’s degree in Mathematics, a master’s degree in Biostatistics and a Ph.D. in Molecular Epidemiology of Infectious Diseases from the NKUA, Greece. Her PhD focused on characterizing the HIV-1 epidemic in Greece using molecular epidemiology methods. The results of her Ph.D. made a significant impact to the better understanding of the dispersal patterns of HIV-1 and HIV-1 resistant strains, as well as the factors associated with the local spread in Greece [160,161].

Evangelia is specialized in the molecular epidemiology of HIV-1, SARS-CoV-2, and Hepatitis viruses. Her research activity focuses on the fields of epidemiology, molecular epidemiology, virus evolution, bioinformatics, and applications of molecular epidemiology to public health. In the context of her research in virology, she has used a collection of bioinformatic analysis tools that helped her analyze large datasets of viral genome sequences. Most of these tools were used for subtyping/classification, alignment, sequence similarity, phylogenetic, phylogeographic and phylodynamic analysis. Specifically, she has been involved in studies related to the investigation of how transmission networks associate with social networks, the estimation of transmission dynamics, dispersal patterns, and spatiotemporal characteristics of viral infections, as well as outbreak investigation [162,163,164,165,166].

Evangelia joined the EVBC in 02/2018. She is regularly reading the newsletter to stay updated and using the tool collection.


**Dr. Neta S. Zuckerman: Discovering and monitoring viruses that threaten public health in Israel.**


Neta S. Zuckerman obtained her BSc, MSc and PhD degrees in computational biology at Bar-Ilan University, Israel, developing computational methods to characterize B-cell receptor evolution and immune responses in B cell-related malignancies and autoimmune diseases. As a post-doctorate at Stanford University, USA, she developed a novel method for gene expression deconvolution and applied it to study the composition, proportions and cell-specific gene expression of cell populations in the setting of breast cancer. At Genentech Inc., USA, as a lead bioinformatics scientist, she developed and applied novel methods and pipelines to research the role of the immune system in various lung-related disorders and autoimmune diseases to understand patient heterogeneity, define bio-markers for patient groups and identify new therapeutic targets.

Currently, Neta heads the Bioinformatics and Genomics center at the National Virology Laboratory, Public Health Services, Israel Ministry of Health. The center specializes in whole genome sequencing of viruses from a myriad of clinical and environmental samples, and researches viruses concerning public health in Israel, including metagenomics virome discovery for diagnostics and environmental monitoring, molecular phylogenetics to characterize viral outbreaks and gene expression studies of viral infections. During COVID, the center plays a pivotal role in SARS-CoV-2 sequencing, variant surveillance and discovery in clinical and sewage specimen in Israel.

Neta joined the EVBC in 08/2018. Neta regularly participates in the annual conferences, where she gave a talk on single cell molecular dynamics of West Nile virus in 2020 [6].


**Dr. Alba Pérez-Cataluña: Monitoring viruses in wastewater.**


Alba Pérez-Cataluña is a postdoctoral researcher at the Institute of Agrochemistry and Food Technology (IATA-CSIC) in Spain. She has a degree in Biology and a doctorate in Biomedicine.

Her studies in virology have been focused on the bioinformatic analysis of the virome present in different types of samples (wastewater, drinking water, feces, …) using massive sequencing techniques [167]. In addition, since the beginning of the pandemic caused by SARS-CoV-2, she has worked on the detection and quantification of this virus by RT-qPCR in wastewater using it as a tool for Wastewater-Based Epidemiology (WE). In these samples she has carried out the sequencing and bioinformatics of SARS-CoV-2 genomes for the analysis of existing variants and the introduction of new ones in the studied populations [168]. Currently, the WBE lessons learned during the pandemic will be applied to the study of other viruses in wastewater, such as other respiratory viruses and West Nile virus.

Alba joined the EVBC in 03/2019. She is attending the annual conference and was part of the organizing committee of ViBioM 2022 [8]. She is also co-editor of the special issue “Virus Bioinformatics 2022” in MDPI Viruses.

### 3.6. Viral Ecology


**Dr. Janina Rahlff: Investigating phage ecology in marine systems.**


Janina Rahlff is a marine biologist and postdoctoral researcher and works at the Linnaeus University in Kalmar, Sweden in the group of Karin Holmfeldt. Janina investigates viral-bacterial interactions in the sea-surface microlayer, which is the uppermost, 1-mm thin layer of the water surface, with recent investigations being carried out in the Baltic Sea and the central Arctic Ocean. Microorganisms and potentially viruses in the microlayer are important in controlling gas exchange and organic matter cycling at the air-sea interface [169,170]. She is particularly interested which bacteriophages exist in different sea surface phenomena, like in microlayer from surface slicks (calm, wave-dampened zones at the sea surface) or floating sea foams, which are already known to be bacterial hotspots [171]. She has isolated highly diverse phages and uses bioinformatics to answer different ecological questions. Bioinformatic investigations allow her to study dispersal patterns [172], to explore viral abundance and diversity, to match viral genomes to their hosts, to investigate adaptive immunity of bacteria against viruses, or to detect auxiliary metabolic genes.

Janina joined the EVBC in 11/2021. She enjoys being part of this vibrant and diverse community, which she recently met during the annual conference in 2022. In July 2022, she will present her research in the ECR Viromics Webinar Series. Also, her member profile was featured in the EVBC Newsletter Vol. 57 (02/2022).


**Dr. Laura Miozzi: Characterizing viral biodiversity.**


Laura Miozzi is a plant bioinformatician at the Institute of Sustainable Plant Protection of the National Research Council of Italy (IPSP-CNR). After the bachelor’s degree in natural sciences, she obtained a PhD on the molecular basis of the interaction between the mycorrhizal fungus *Tuber borchii* and its host plant *Cistus* [173].

In 2004, she started working as bioinformatician in the computational biology group at the Department of Genetics, Biology and Biochemistry, University of Torino. Her activity was mainly focused on the development of data-mining approaches for gene functional annotation, based on expression data [174]. Since 2006 she has been a researcher at the IPSP-CNR where she integrates bioinformatics and molecular biology approaches. Her research activity focuses on three fundamental lines: the study of plant-virus interaction [175,176,177,178], the characterization of viral biodiversity and viral etiological agents [179,180,181], and the development of environmentally friendly approaches for plant protection [182].

Laura joined the EVBC in 08/2020. She is regularly reading the newsletter and actively attending the lectures and the annual conference. She often uses the tool collection as a precious reference for her work.

## 4. Conclusions

The EVBC is bringing together the excellence of virology and bioinformatics for more than five years now. We are constantly trying to develop our services and activities to be of valuable help to the research community, to provide numerous opportunities for scientific exchange and collaboration, and to increase the international visibility of virus bioinformatics. With all our activities we are seeking to provide a supportive environment for talented researchers that is free of any bias, be it gender, ethnicity, nationality, disability, or LGBTQ. Our female members at various career stages have made outstanding scientific achievements, such as discovering novel and ancient viruses, developing vaccines and therapeutics, understanding viral transmission dynamics, and many more. The female scientists introduced here are enriching our community with their excellent research and engagement in the EVBC activities. In the future, we would like to expand our efforts beyond gender discrimination [183]. We are planning to reach out to our community to raise awareness and to discuss how EVBC can prevent discrimination and provide support.

If you think that the aims and services of the EVBC could also support your research, we would be delighted if you are interested in joining and enriching our community. To become a member, please fill in the application form (http://evbc.uni-jena.de/membership-application/).

## Figures and Tables

**Figure 1 viruses-14-01522-f001:**
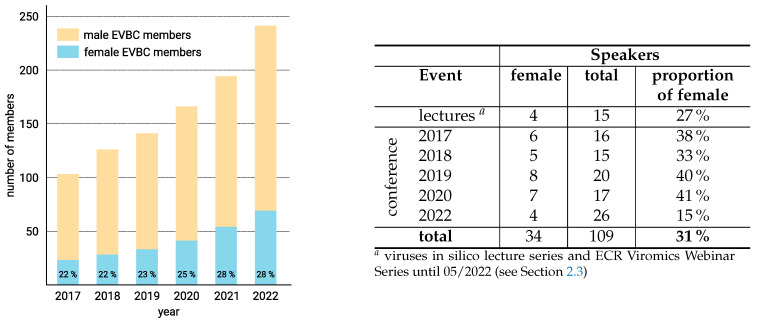
Female scientists in the EVBC. (**Left**) Number of members in the EVBC. Since its foundation, we have succeeded in steadily increasing the proportion of female members. (**Right**) The EVBC seeks to ensure adequate participation of female speakers in lectures and conferences.

**Table 1 viruses-14-01522-t001:** EVBC services and activities. The EVBC is engaging in several activities to solidify the exchange of ideas, initiate scientific cooperation between bioinformaticians and virologists and increase the international visibility of virus bioinformatics. If you are interested in joining the EVBC, please fill in the application form (http://evbc.uni-jena.de/membership-application/).

Service/Activity	See Sec.	Description	Link
monthly newsletter	Section 2.1	recent research results, upcoming events, job vacancies and further announcements	https://evbc.uni-jena.de/evbc-newsletter/
EVBC Twitter	Section 2.1	sharing recent publications, vacancies and event announcements	https://twitter.com/EVirusBioinfC
publication radar	Section 2.1	monitoring the publications of our members and highlighting all virus bioinformatics related papers in the newsletter and on Twitter	https://evbc.uni-jena.de/publications/
calendar	Section 2.1	listing conferences, workshops, lectures etc. (including submission and registrations deadlines)	https://evbc.uni-jena.de/events/
vacancies	Section 2.1	collection of vacancies offered by our members	https://evbc.uni-jena.de/vacancies/
annual conference on virus bioinformatics	Section 2.2	exchange the latest research results and experiences with an extensive network of leading experts and aspiring young scientists	https://evbc.uni-jena.de/events/vibiom/
viruses in silico lecture series	Section 2.3	keep you up to date with the latest developments in virus bioinformatics, especially new tools that might help you in your research	https://evbc.uni-jena.de/events/viruses-in-silico/
ECR Viromics Webinar Series	Section 2.3	aimed at early career researchers studying viruses in complex communities	https://evbc.uni-jena.de/events/ecr-viromics-webinar-series/
workshops	Section 2.4	laying the foundation for a high quality education in virus bioinformatics	https://evbc.uni-jena.de/events/workshops/
tool collection	Section 2.5	curated collection of virus bioinformatics tools	http://bit.ly/evbctools
collaborative research projects	Section 2.6	implementing jointly funded projects that achieve more than the sum of their parts	https://viroinf.eu/
special issues	Section 2.7	recurring special issue on virus bioinformatics	https://evbc.uni-jena.de/special-issues/

**Table 2 viruses-14-01522-t002:** History of the annual conference of the EVBC. The annual conference of the EVBC offers the opportunity to exchange the latest research results and experiences with an extensive network of leading experts and aspiring young scientists in both areas and engage in inspiring discussions.

Name	Abbr.	Date	Location	# Part.	Key Outcomes	Report
1st Meeting of the European Virus Bioinformatics Center		6–8 March 2017	Jena, Germany	∼100	Discussion of the role of the EVBCFounding of the EVBCElection of the first Board of DirectorsInsights into EU policy and funding opportunities	
2nd Annual Meeting of the European Virus Bioinformatics Center		9–10 April 2018	Utrecht, The Netherlands	∼120	Extension of the EVBC network to include America and Asia Discussion and design of joint projects	[7]
3rd Annual Meeting of the European Virus Bioinformatics Center		28–29 March 2019	Glasgow, UK	∼110	Inclusion of contributed talks to the scientific programAwards for junior scientists	[5]
International Virus Bioinformatics Meeting 2020	IVBM 2020		Bern, Switzerland/virtually	∼120	Renaming of the conferenceOnline format due to pandemicElection of Board of DirectorsPresentation of VIROINF network (see Section 2.6)	[6]
International Virus Bioinformatics Meeting 2022	ViBioM 2022		Valencia, Spain/virtually	100–150 (380 ^*a*^)	Satellite meeting on SARS-CoV-2“Ask me anything” with the keynote speakersVirtual poster session in individual breakout rooms	[8]

^*a*^ Number of registrations.

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
