# Peer review of "Women in the European Virus Bioinformatics Center"

_viruses, 2022, doi:10.3390/v14071522_

Round 1

Reviewer 1 Report

This is a terrific summary of the many efforts by the EVBC and a wonderful highlight of the fantastic work that members of the EVBC perform. I very strongly support and encourage its publication. 

I saw the link at the bottom of the manuscript on how to join the EVBC but encourage you to add that earlier in the paper (or even in the abstract?) for people with shorter attention spans!

On line 106 how many registered participants where there?

One of the things that I try to encourage is more inclusion of women speakers at conferences. I could not easily identify a way to filter the EVBC member list on the website to just women scientists, and perhaps that could be added and highlighted in this paper?

Author Response

We thank the reviewer for this very positive evaluation of our manuscript and the valuable comments for improvement. Please see the detailed comments in the attached PDF.

Reviewer 2 Report

Brief Summary

Hufsky et al. embark on a journey to highlight outstanding scientists in the European Virus Bioinformatics Center. Specifically, they discuss the importance of developing new bioinformatics tool specifically for virology. In parallel, Hufsky et al. introduce the cutting-edge research done by the women of the EVBC center. 

Significance

The emergence of new sequencing technologies has generated “big data” in the fields of virology, creating an urgent need for virology specific bioinformatics tools. Yet, existing biases and discrimination in the scientific community, and specifically in virology and data analysis, have been pushing brilliant scientists from historically excluded groups out of academia. Here, the authors spotlight outstanding women from the EVBC center. Highlighting their contributions to science emphasizes that diversity, equity, and inclusion are essential to the mission of the Viruses journal and the virology scientific community. Furthermore, celebrating the science done by the women of the EVBC center is an important step towards supporting these remarkable individual researchers and is crucial for fostering future scholars by featuring role models for younger scientists. 

Recommendations: 

I enthusiastically recommend accepting this paper with minor revisions for publication at the Viruses Journal. I am listing below minor suggestions for clarifying details described in this commentary.   

Comments: 

Line 169: “Despite bringing together scientists during conferences, workshops and seminars to promote scientific exchange, the EVBC is initiating large collaborative research projects.” This sentence is not clear. Perhaps the authors meant to say, “As a direct result of bringing…”, or “In addition to bringing… “. Please rephrase this sentence. 

Lines 108: “The gender gap in science persists to a greater degree than in other professions, but in particular in high-end, math-intensive fields such as computer science and engineering.” Please add references to this paragraph. Such as the following EU, USA, and Israeli reports on gender gap in academia/science: 

-       She Figures, EU report on Gender in Research and Innovation Statistics and Indicators, 2021. https://op.europa.eu/en/web/eu-law-and-publications/publication-detail/-/publication/67d5a207-4da1-11ec-91ac-01aa75ed71a1

-       Women in STEM, 2017 update of the economics and statistics administration office of the chief economist, the US department of commerce. https://www.commerce.gov/sites/default/files/migrated/reports/women-in-stem-2017-update.pdf

-       Women Representation In Academia, 2018 report of the Knesset research and information center, the Israeli committee for promoting women and gender equality. [In Hebrew]. https://fs.knesset.gov.il/globaldocs/MMM/64a5fee7-6462-e811-80dd-00155d0a0b8d/2_64a5fee7-6462-e811-80dd-00155d0a0b8d_11_10507.pdf

Please add references to gender gap in academia, such as: 

-       Progress toward gender equality in the United States has slowed or stalled, England et al. 2020, PNAS.

-       Science faculty’s subtle gender biases favor male students, Moss-Racusin et al. 2012, PNAS.

-       Meta-analysis of field experiments shows no change in racial discrimination in hiring over time, Quillian et al. 2017, PNAS.

-       Discrimination against Queer Women in the U.S. Workforce: A Résumé Audit Study, Mishel, 2016, Socius.

Lines 200-206: “The more women get involved in science – or any other traditionally male-dominated 200 field – the greater the overall knowledge in that field becomes.” Please add references to this paragraph. Such as:

-       Mathias Wullum Nielsen [email protected], Sharla Alegria, Love Börjeson, Henry Etzkowitz, Holly J. Falk-Krzesinski, Aparna Joshi, Erin Leahey, Laurel Smith-Doerr, Anita Williams Woolley, and Londa Schiebinger, Gender diversity leads to better science, 2017, PNAS, https://doi.org/10.1073/pnas.1700616114

-       Julia King, Benefits of Women in Science, 2005, Science

Lines 207-218:” As an example, for years, animals used in scientific experiments were predominantly male 207 so as not to falsify experimental results with female hormone fluctuations. …” Please add references to this paragraph. Such as: 

-       Zucker, I., Beery, A. Males still dominate animal studies. Nature 465, 690 (2010). https://doi.org/10.1038/465690a

-       Shansky R. M, Are hormones a “female problem” for animal research? Outdated gender stereotypes are influencing experimental design in laboratory animals SCIENCE, 31 May 2019, Vol 364, Issue 6443, pp. 825-826, DOI: 10.1126/science.aaw7570 

-       Annaliese K.Beery, Inclusion of females does not increase variability in rodent research studies, Current Opinion in Behavioral Sciences, Volume 23, October 2018, Pages 143-149

Lines 216-221: “To keep more talented women in research, we need to ensure that they are not kept at lower levels but are given the opportunity to lead the field. … ” Please add references to this paragraph, supporting claims about the importance of role models in inclusion, retention, and promotion of women in science. Please refer to future efforts to include LGBTQIA, disability, and BIPOC communities in the EVBC center. Will there be better conditions for maternal and parental leaves? Is there any childcare support from the EVBC center? Are you creating new funding opportunities focused on promoting scientists from historically excluded communities? More literature on this subject: 

-       What Black scientists want from colleagues and their institutions, Frustrated and exhausted by systemic racism in the science community, Black researchers outline steps for action. 22 June 2020, nature

-       Opinion: In the wake of COVID-19, academia needs new solutions to ensure gender equity, Malisch et al. 2020, PNAS.

-       Virginia Gewin, Women can benefit from female-led networks, Support-based peer associations offer professional value, 20 December 2018, Nature

Line 236: “In addition, she is involved in several (mainly German) outreach activities on virus bioinformatics.” Please name which outreach activity/organization, or rephrase to “ she is passionate about science outreach activities on the subject of virus bioinformatics”. 

Please change the first line in each highlight to “Dr. full name”. such as line 274, “ Sofia is a biologist…” to “Dr. Sofia Paraskevopoulou is a biologist … “. And “Dr. last name, in the rest of the highlight, such as line 288 ”Sofia is now…” to Dr. Paraskevopoulou is now”. 

Please consider adding photos of the highlighted researchers. 

Author Response

We thank the reviewer for this very positive evaluation of our manuscript and the valuable comments for improvement. In particular, we thank for the many valuable and interesting publications s*he pointed out to us. Please find the detailed comments in the attached PDF.
